# Interlayer Hybridization of Virgin Carbon, Recycled Carbon and Natural Fiber Laminates

**DOI:** 10.3390/ma13214955

**Published:** 2020-11-04

**Authors:** Peter R. Wilson, Alon Ratner, Gary Stocker, Frank Syred, Kerry Kirwan, Stuart R. Coles

**Affiliations:** 1WMG, University of Warwick, Gibbet Hill Road, Coventry CV4 7AL, UK; p.wilson.5@warwick.ac.uk (P.R.W.); a.ratner@warwick.ac.uk (A.R.); g.stocker@warwick.ac.uk (G.S.); kerry.kirwan@warwick.ac.uk (K.K.); 2Brüel and Kjær Global Engineering Services, Millbrook, Bedfordshire MK45 2YT, UK; frank.syred@bksv.com

**Keywords:** hybrid composites, internal friction/damping, mechanical testing, vacuum infusion

## Abstract

To meet sustainability objectives in the transport sector, natural fiber (NF) and recycled carbon fiber (RCF) have been developed, although they have been typically limited to low to medium performance components. This work has considered the effect of interlayer hybridization of woven NF and non-woven RCF with woven virgin carbon fibers (VCF) on the mechanical and damping performance of hybrid laminates, produced using double bag vacuum infusion (DBVI). The mean damping ratio of the pure laminates showed a trend of NF>RCF>VCF, which was inversely proportional to their modulus. The tensile, flexural and damping properties of hybrid laminates were dominated by the outermost ply. The VCF-RCF and VCF-NF hybrid laminates showed a comparatively greater mean damping ratio. The results of this work demonstrate a method for the uptake of alternative materials with a minimal impact on the mechanical properties and improved damping performance.

## 1. Introduction

Spurred by an increasingly global consensus on sustainability targets, the transport sector has focused on lightweighting as a means of reducing vehicle emissions. Product design with lightweight materials has resulted in considerable growth in demand for fiber-reinforced plastics. For example, the global demand for carbon fiber by automotive industries reached USD 2.4 billion in 2015 and has been projected to reach USD 6.3 billion in 2021 [1]. However, while lightweighting has been a key part of efforts to improve the sustainability of the transportation sector, the production of large volumes of synthetic fiber-reinforced plastics has presented environmental and legislative challenges for dealing with industrial waste that is typically disposed of by landfill. Consequently, reinforcements such as sustainably sourced natural fibers and recycled carbon fibers have been investigated as a means of reducing the environmental impact of end-of-life disposal and enabling closed-loop recycling circularity in the composites industry [2,3,4,5].

Given the relatively high embodied energy of virgin carbon fiber 183–286 MJ/kg [6] and glass fiber 13–54.7 MJ/kg [6,7] reinforcements, natural fibers 3.64–86 MJ/kg depending on processing [4,8] have been investigated as lower cost and less environmentally impacting materials for automotive applications [8,9,10,11]. Recycling of carbon fiber composites has been shown as an economically viable means of achieving a circular economy for carbon fiber-reinforced plastics in the composites industry [12,13,14,15,16]. Recycled carbon fibers have been commercialized, in one instance, by ELG Carbon Fiber as Carbiso^TM^ non-woven mats. ELG Carbon Fiber claim that their recycled carbon fibers retain 90% of the tensile strength compared to the original strength and that the cost of recycled carbon fibers is substantially lower than that of virgin feedstock [17,18]. Due to the discontinuity of the fibers and reduced axial alignment, non-woven fabrics are characterized by comparatively lower planar tensile strength and stiffness than fabrics woven with the same filaments [15].

Fibrous networks are also recognized for their vibration absorbing properties [19], and natural fiber reinforcements are valued as sustainable materials with especially favorable damping properties [20,21,22,23,24]. The damping ratio of recycled carbon fibers has been investigated [2]; however, to the authors’ knowledge, it has not been previously reported for non-woven recycled carbon fiber laminates. 

Hybridization has been implemented as a means of synergizing the properties of different fiber types—for instance, by combining the high tensile strength and stiffness of carbon fibers with the lower cost and reasonable energy absorption of natural fiber reinforcements [25,26,27,28,29,30,31]. Hybridization of recycled carbon and virgin fibers has recently been proposed as a research gap to enable high quality materials [32]. Research that has been undertaken on hybridization includes the interlayering of carbon fiber and natural fiber woven fabrics [33] and recycled carbon fiber and natural fiber [34,35]. The resulting hybrid laminates gain a combination of properties from the constituent reinforcements, including tensile and flexural strength and stiffness, impact absorption, vibration damping ratio and density [36,37,38,39]. 

For instance, the Young’s modulus of hybrid carbon and flax fiber laminates has been observed to be proportional to the fiber volume fraction of carbon fibers while the damping ratio increases in proportion to the fiber volume fraction of flax fibers [40,41,42]. Factors influencing the relationship between a given physical property and the relative fiber volume fraction of its dissimilar reinforcements include the fiber aspect ratio, fiber orientation, stacking sequence and material properties of the dissimilar laminae [43,44,45,46,47].

The numerical synergy of these physical properties with the fiber volume fraction in interlayered hybrids is non-linear and the outermost plies have been observed to dominate flexural stiffness and vibration damping [48,49]. This is consistent with classical plate theory; since bending stresses are maximal in layers furthest from the neutral axis, flexural stiffness would be expected to be maximal in hybrid laminates with stiff reinforcing plies located in the outermost layers [50]. Similarly, the damping ratio of a hybrid laminate may be expected to be most highly influenced by the outermost layers when subjected to flexural stress waves that propagate from the exterior plies of a laminate through its thickness. A hybridization effect also occurs in tension with sandwich laminates that contain reinforcements of different extensibility. As explained by Kretsis [51], the ultimate tensile strength of a composite can be increased by encasing brittle reinforcement plies with more extensible phases that permit continued stress transfer after the yield strength of the reinforcement has been exceeded in tension. In this manner, the tensile strength of a hybrid laminate containing both strong and weak plies can be greater than that predicted by the rule of mixtures. Song observed slightly higher tensile modulus and strength in carbon fiber and aramid fiber sandwich laminates in which the stronger carbon fiber plies were located at the center of the laminate and deduced that tensile stress was concentrated towards the center of tensile coupons of sandwich laminates [52]. In a study of aramid, glass and carbon fiber hybrid laminates, Wang confirmed that flexural and damping properties were dominated by the stiffness of the outer ply and noted a relatively minor effect of stacking sequence on tensile properties [53].

The deviation of an observed property from a linear relationship with the volume fraction of a given constituent was termed by Marom et al. as either ‘positive’ or ‘negative’ if the magnitude of the observed property was either higher or lower than the value predicted by the rule of mixtures [54]. This terminology will be used in this publication for convenience of expression.

While hybridization of sustainable materials with traditional reinforcements has been demonstrated, development of new hybrid composites that utilize RCF is required to promote the utilization of these materials into a wider range of promising structural applications. This study examines the performance of pure and hybridized laminates, containing two reinforcements from virgin woven carbon fiber, recycled non-woven carbon fiber and natural fibers, manufactured via a double bag vacuum infusion (DBVI) technique.

The damping, tensile and flexural properties will be compared against resulting fiber volume fraction and void content, which have not been previously reported for these hybrid laminates.

## 2. Materials and Methods 

### 2.1. Materials

All panels were manufactured by double bag vacuum infusion (DBVI) using IN2 epoxy infusion resin with AT30 hardener (Easy Composites Ltd., Stoke-on-Trent, UK) and mixed 100:30 by weight. Toray T300 carbon fiber 200 gsm 2/2 twill fabric (Sigmatex Ltd., Runcorn, UK), Carbiso M SM45D recycled carbon 200 gsm non-woven fiber fabric (ELG Carbon Fiber Ltd., Bilston, UK) and Composites Evolution Biotex flax fiber 200 gsm 2/2 twill fabric (Easy Composites Ltd., Stoke-on-Trent, UK) were used throughout this work.

There are a greater number of filaments aligned transverse to the roll direction in the Carbiso M SM45D non-woven fabric; consequently, the alignment of the fabric transverse to the roll direction was defined as 0° for the purpose of defining the lay-up sequence. The remaining woven Biotex flax and Toray T300 fabrics are balanced and were defined as 0° along the fabric roll and 90° transverse to the roll direction. Flax fabrics were dried overnight in an oven at 50 °C before infusion.

‘Pure’ laminates containing a single type of fiber reinforcement and ‘hybrid’ laminates containing two types of fiber reinforcement are shown in Figure 1 and Table 1. The mass and thickness of each panel are given in Table 1. Due to differences in the thickness and the wettability of dry fabrics, variation was observed in the mass of the cured panels. The lay-up sequences were chosen in an effort to manufacture balanced laminates and to obtain similarity in the cured thickness of the panels. However, there was some variation in the thickness of the panels due to the dissimilarity of ply thicknesses for the reinforcement fabrics that were available at the time of manufacture and the different rates of resin uptake by the various dry fiber types. This is reflected in the variation of the thickness of the panels shown in Table 1.

### 2.2. Double Bag Vacuum Infusion

A variant on resin infusion under flexible tooling called double bag vacuum infusion (DBVI) [55,56,57] was used for all laminates; a diagram of the DBVI setup is shown in Figure 2. Dry reinforcement fabrics cut to 500 × 500 mm were laid onto a steel caul plate covered with a polytetrafluoroethylene (PTFE) coated glass cloth to aid mold release. The fabric was overlaid with nylon peel ply and infusion mesh with infusion connector ports and infusion spiral tubing on either side of the fabric to distribute vacuum pressure across the width of the caul plate in order to generate a rectilinear flow front and then sealed in an inner nylon vacuum bag. A vacuum was drawn from the inner bag consolidating reinforcements and consumables. This vacuumed assembly was then sealed in a secondary vacuum bag around the infusion pipes and an additional layer of breather cloth was utilized to distribute vacuum. The same vacuum pump was utilized for the outer bag to provide vacuum consolidation pressure throughout cure. IN2 resin and AT30 hardener were weighed in paper cups and mixed by hand using a wooden stick until a homogenous mixture was produced. The ports were connected with polyethylene tubes interconnected with infusion values and into a well-mixed pot of resin on the inlet side and a vacuum pump with a catch pot for excess resin on the outlet side. All consumables were obtained from Tygavac Ltd. (Chadderton, UK).

When the inlet and outlet ports were opened, resin was drawn through the inlet tube and permeated the dry fabric. The infusion valves were closed once resin began to collect in the catch pot. Vacuum pressure was maintained on the secondary bag throughout curing. As the IN2 resin system optimally cures at room temperature, panels were left undisturbed for 24 h before being released from the caul plate. In addition, all panels were post-cured in a convection oven at 80 °C for 4 h, following the profile suggested by the resin datasheet.

### 2.3. Experimental Modal Analysis Method

Cured laminates were trimmed to 420 mm × 420 mm and experimental modal analysis was undertaken on these panels. Experimental modal analysis was undertaken on unrestrained panels using impulse excitation at Brüel and Kjær Sound and Vibration Engineering Services (Nærum, Denmark). The panels were suspended on bungee cords, which were loosely tied between two equally spaced steel rods that were clamped to a rigid workbench, as shown in Figure 3.

This method was used to represent an idealized unrestrained (free-free) condition by ensuring that the rigid body modes of the panel on the suspension system were less than 10% of the lowest flexural resonant frequencies of the test panels. Impulses were generated at the location shown in Figure 3 using a Brüel and Kjær 8206 Impact Hammer (Nærum, Denmark) fitted with a polyacetal tip to ensure that the input force spectrum adequately covered the frequency range from 5 to 1000 Hz without introducing excessive noise to the frequency response function. The vibration response on the test panel was measured using a Brüel and Kjær 4518 uniaxial accelerometer (Nærum, Denmark), which was placed at least 75 mm away from the edges of the panels and 50 mm away from the underlying bungee cords to reduce the influence of the suspension system. This location also avoided axes of symmetry along the square panel that might coincide with geometrical nodes and anti-nodes of resonance. The accelerometer is contained within a miniaturized titanium housing and has a mass of 1.45 g. The size and mass of the miniature accelerometer is considered to be insignificant with respect to the mass and dimensions of the panel and therefore has been assumed not to significantly bias the modal response of the panel during the tests. The accelerometer was bonded to strips of Kapton tape using cyanoacrylate adhesive to avoid ingress of the adhesive into the laminates, preventing the adhesive from influencing the material properties of the panels. A Brüel and Kjær LAN-XI data acquisition system (Nærum, Denmark) was used for the measurements.

Frequency response functions (FRF) between the impact hammer force and the resultant panel vibration were generated for each panel by averaging the responses obtained from several individual impacts. An exponential time domain window was applied to minimize integration leakage in the spectral calculation. The FRFs were measured between 0 and 1600 Hz, with a frequency resolution of 0.25 Hz. Since mode shapes cannot be determined from measurement at a single position [58], curve fitting modal analysis software was used to estimate the frequency and damping ratio from the FRFs for the flexural panel modes up to 1000 Hz.

Rigid body motions of the panels which occurred below 50 Hz were excluded from this analysis. All the measurements and the FRF curve fitting analysis were conducted using Brüel and Kjær Connect Modal Analysis 8420 software (Nærum, Denmark).

### 2.4. Mechanical and Optical Sample Preparation

Post-modal analysis panels were further cut into tensile and three-point bend coupons as well as optical microscopy samples, as shown in Figure 4. All cutting was undertaken in the direction of laminate infusion so that the cross-sections of all coupons and optical microscopy samples were perpendicular to the direction of laminate infusion. For laminates containing RCF, they were tested along the principal direction. In this manner, a total of nine optical microscopy samples were obtained from the center, edges and corners of each panel, as highlighted with red circles in Figure 4.

Tensile samples were tabbed with strips of FR-4 fiberglass using Araldite Standard epoxy prior to cutting tensile coupons while three-point bend coupons and samples for optical microscopy were directly cut from the panels. The mean width and thickness of each coupon were measured at three points along the gauge length with Vernier calipers. A total of 6 tensile coupons and 10 three-point bend coupons were prepared from each sample. 

Optical microscopy samples were mounted using plastic retainer clips so that the samples were polished perpendicular to the direction of infusion. EpoFix resin and hardener (Struers Ltd., Rotherham, UK) was mixed using a ratio of 100:12 and samples were cold mounted under vacuum and were degassed for 10 min to remove bubbles. Samples were then polished using a Buehler AutoMet 300 Pro (Buehler, Lake County, IL, USA) using a P400 SiC abrasive, a 9-mm diamond polish, a 6-mm diamond polish and a 0.05-mm diamond polish. Polished samples were then stored in lint-free cloths.

### 2.5. Optical Microscopy

Optical images were taken using a Zeiss AX10 optical microscope (Carl Zeiss AG, Oberkochen, Germany) with an automatic stage Axiocam 305 camera using a 10X optical zoom, resulting a resolution of 1 pixel to 2.97 µm. The image stitching application in Zeiss ZEN core version 2.6 was used for optical image acquisition. Each constituent of a laminate was analyzed using thresholding function in ImageJ Version 1.52 P. The region of interest around each constituent material in a laminate was manually selected using the polygon tool prior to threshold volume percentage analysis. To enable consistency between all image sets, an automatic thresholding was used. The volume fraction of carbon fibers was determined using the mean automatic thresholding and the void fraction was determined using the Auto Max entropy threshold. Since the pixel intensity associated with natural fibers overlapped with that of voids, both natural fibers and voids were initially selected using an automatic mean and then voids were manually subtracted to distinguish between the fibers and void volume fractions in specimens containing natural fibers. The 9 specimens per laminate type were averaged and a standard deviation was determined.

### 2.6. Tensile Testing

Tensile tests were undertaken with guidance from testing standard ISO 527-4-1997 [59] using Type 2 specimens. Testing was undertaken on an Instron 5800R (Norwood, MA, USA) universal test frame with wedge grips at a quasi-static crosshead speed of 1 mm/min. Load was measured using an Instron 100 kN load cell.

#### 2.6.1. 3D Stereo Digital Image Correlation (DIC)

Strain was determined from digital image correlation using a GOM ARAMIS 12 M Adjustable camera system. Cameras were levelled and aligned to the samples’ surface using inbuilt spirit level and rulers. Load data were converted into a voltage signal that was exported to Instron Bluehill 2 testing software. This load voltage was added as an analogue input into Aramis Professional (GOM Ltd., Coventry, UK) for synchronized load logging with image capture. The details of the digital image correlation (DIC) instrumentation, setup parameters are given Table 2.

#### 2.6.2. Image Correlation Analysis

Aramis Professional version 2018 (GOM Ltd., Coventry, UK) DIC software was used to determine the surface displacements and resulting axial and transverse strains. Image correlation was performed using the least squares matching method with automatic seeding point determination from Aramis professional. A mask was inset around the sample edges to account for edge effects. Samples were aligned using Aramis 3.2.1 alignment. The X and Y strains were exported against the logged load data and converted into stress. Stress and X and Y strain were calculated into tensile modulus and Poisson’s ratio using Origin 2018 according to ISO 527-1-2012 [60].

### 2.7. Three-Point Bend Testing

Three-point bend tests were undertaken with guidance from testing standard ISO 14125-1998. Coupons were prepared according to dimensions for Class II materials. Testing was undertaken on an Instron 3360 universal test frame using a flexural testing fixture with rollers that matched the diameters stated in the standard at a quasi-static crosshead speed of 1 mm/min. Load was measured using an Instron 2580 (Norwood, MA, USA) 10 kN load cell and deflection was determined with an extensometer in contact with the mid-point of the coupons. Flexural stress, strain and modulus were calculated following Method A in ISO 14125-1998 [61].

## 3. Results and Discussion

### 3.1. Optical Microscopy: Fiber Volume Fraction, Void Content 

Representative optical micrographs for each laminate are given in Figure 5. The pure laminates shown in Figure 5a–c all show macroscopic voids towards the top surface of the laminate—this could be explained due to buoyancy effects prior to curing. Hybrid laminates exhibited a resin-rich layer at the interface between constituent layers that was not observed in the pure laminates. An explanation for this is due to the disorder in geometrical packing caused by the combination of differently shaped fiber architectures, as was observed by Bachmann et al. in hybrid flax and recycled carbon fiber laminates [35]. While this resin-rich layer might represent an unwanted parasitic mass in the context of lightweighting, the authors speculate that it might offer a mechanism for increasing interlaminar shear strength by blunting crack propagation between laminae. This could be a beneficial feature for increasing impact resistance, although this would have to be investigated as part of further work.

The mean fiber volume fractions of each laminate by constituent are given in Figure 6a. The mean fiber volume fractions (V_f_) for the pure laminates are 48.9%, 17% and 32% for VCF, RCF and NF laminates, respectively. The larger V_f_ for woven carbon and flax relates to the tighter packing density of their woven structures, while the RCF fabrics have a lower packing density and discontinuous fibers, resulting in a lower V_f_. The NF fabric is made from low-twist yarns without continuous fibers, which minimizes the fiber-packing density, resulting in a lower V_f_ compared to VCF laminates.

The structures of each hybrid laminate represent a synthesis of the morphological features inherent to each constituent, which resulted in an averaging of the V_f_ for each hybrid laminate, while the V_f_ of each constituent layer was unaffected. As seen in Figure 6b, there is a negligible change in the mean V_f_ of each constituent layer when hybridized with other materials.

The void content of the laminates by constituent is given in Figure 7a. The mean void content for the pure laminates are 0.8%, 1.3% and 2% for VCF, RCF and NF laminates, respectively. The minimal void content for the pure VCF laminate is expected due to the compatibility of the epoxy resin with the sizing on the carbon fiber filaments and the relatively smooth flow front of resin by capillary action along the continuous tows. The comparatively higher void content for the pure RCF laminate can be accounted for by the semi-random fiber-packing and discontinuous fibers of the recycled fabric that implies greater tortuosity for resin infusion, which may be less permeable to the flow of resin and thus contribute to the formation of a greater volume of voids. In a similar manner, the low-twist discontinuous yarns of the NF woven fabric offer closed cavities which may act as nucleation points for the formation of voids. Furthermore, the hollow lumen structure, relatively large fiber diameter and inhomogeneity of the surface of the natural fiber filaments, as well as poorer interfacial compatibility [62] of natural fiber fabrics with the infusion resin, may have contributed to a higher void content for laminates containing natural fibers than those with carbon fiber reinforcements.

The void contents of the hybrid laminates shown in Figure 7b demonstrate a similar trend to the V_f_ with an apparent averaging between each of the constituents. This apparent averaging of the void content, shown in Figure 7, suggests a mechanism of migration of voids from constituents with a higher void content to constituents with a lower void content. A possible explanation may be that the pressure gradient caused by the difference in the partial pressure of voids between the constituent layers increased the rate of diffusion of voids from the high-void-content-generating innermost plies towards the outermost plies.

### 3.2. Experimental Modal Analysis

At 1.21%, 0.88% and 0.41%, the respective mean damping ratios of the first ten modes of resonance for pure natural, recycled carbon and carbon fiber laminates are inversely proportional to the modulus of the reinforcing fabric, as shown Figure 8. This is consistent with the expectation that the damping ratio should be inversely proportional to the stiffness of the panel. Specifically, panels with a greater volume fraction of stiffer fibers and weaves with planar fiber alignment are characterized by higher structural rigidity and therefore lower damping ratios. The mean damping ratios for the hybrid laminates are characterized by a negative synergy that is dominated by the material of the outermost layer and this trend is consistent with that observed for carbon–flax hybrids with stiffer plies on the outermost layers.

### 3.3. Mechanical Testing

#### 3.3.1. Tensile Properties

As shown in Figure 9, the Young’s modulus and tensile strength of pure laminates were proportional to the reinforcement stiffness and respective laminate V_f_. The highest values were recorded for VCF, followed by RCF and NF.

For the hybrid laminates, a weak positive synergy can be observed in the Young’s modulus and tensile strength. VCF-RCF and VCF-NF decreased in Young’s modulus by 38.7% and 40.5%, respectively, with respect to pure VCF. The decrease in tensile strength was 40.5% and 47.1% for the VCF-RCF and VCF-NF laminates, respectively, with respect to pure VCF. RCF-NF exhibited a 1.7 and 10.1% decrease with respect to pure RCF in mean Young’s modulus and tensile strength, respectively. This may be explained by the location of the stiffer and stronger carbon fiber plies on the exterior of the hybrid laminates, since, as observed by Song and Wang, the tensile strength and stiffness of a hybrid laminate is minimal when stronger plies are located at the exterior of the lay-up sequence [52,53]. It could be expected that differences in surface chemistry would also contribute to a reduction in tensile strength. However, RCF is a dry fiber waste that has retained its sizing and the sizing was also epoxy-based (as with VCF). Therefore, it would be expected that if incompatibility between fibers and resins was the dominant factor, then both NF hybrid composites would experience a reduction in tensile strength, which is not the case, as shown in Figure 9.

The in-plane Poisson’s ratio of pure laminates, shown in Figure 9, is dependent on the laminate stiffness and fiber geometry. The Poisson’s ratios of VCF and NF are inversely proportional to the stiffness of the woven fabrics, while the non-woven RCF has a higher Poisson’s ratio than might be expected for a woven fabric of corresponding stiffness. A significant feature of the RCF non-woven fabric is the semi-aligned discontinuous fibers which produce an unbalanced ply with some through thickness reinforcement, as seen Figure 6b. This implies that there is less resistance in this transverse direction to biaxial deformation, which may have contributed to a comparatively high Poisson’s ratio. The Poisson’s ratios for VCF-RCF and VCF-NF laminates show strong negative synergy and are dominated by the external VCF ply. By contrast, the Poisson’s ratio of the RCF-NF hybrid laminate is greater than that of the pure laminates for either of its constituents. This suggests a strong positive synergy and an additional mechanism for increasing Poisson’s ratio that may have arisen from a combination of the low transverse biaxial stiffness of the exterior RCF ply and the comparatively high compliance of the underlying NF ply.

#### 3.3.2. Flexural Properties

The flexural modulus and strength of the pure and hybrid laminates are shown in Figure 10. In a similar manner to tensile strength and Young’s modulus, the flexural strength and modulus of pure laminates were highest for VCF, followed by RCF and NF. Strong positive synergy was observed for the hybrid laminates, which is correlated with the strength and stiffness of the fiber reinforcements in the outermost ply. VCF-RCF and VCF-NF saw a 16.5% and 13.3% decrease in mean flexural modulus and a 27.7% and 28% decrease in flexural strength when compared to pure VCF flexural modulus and strength, respectively. RCF-NF exhibited a 1.8% and 1.6% decrease with respect to pure RCF in mean flexural modulus and strength. This trend is consistent with expectations, since bending stresses are maximal in layers farthest from the neutral axis, and in all the hybrids, the stiffest plies were located in the outermost layers [35].

## 4. Conclusions

This work examined the mechanical and damping response of six pure and hybrid laminates manufactured by vacuum-assisted resin infusion to investigate the effect of interlayer hybridization on laminates containing VCF, RCF and NF reinforcements.

V_f_ increased in a trend of VCF > NF > RCF, which can be explained by the lower packing densities and discontinuity of fibers in RCF and NF with respect to VCF. The void content followed a trend of NF > RCF > VCF, which can also be explained by fiber discontinuity and a relatively low level of fiber alignment in RCF, while NF is also characterized by discontinuous fibers and a poor interfacial compatibility between the fibers and the matrix. The structure of each hybrid laminate represents a synthesis of the morphological features inherent to each constituent material present in the laminate, which resulted in an averaging of the V_f_ for each hybrid laminate, while the V_f_ of each constituent layer was unaffected. The void content of the hybrid laminates showed a similar trend to the V_f_ with an apparent averaging between each of the constituents.

The Young’s modulus and tensile strength of laminates were proportional to the tensile strength and stiffness of the constituent layers and a weak positive synergy was observed in hybrid laminates that reflects a correlation with the stiffness of the outermost plies. The strong negative synergy observed for Poisson’s ratio and strong positive synergy for flexural strength and stiffness were also indicative of a strong dependence on the material properties of the reinforcement of the outermost ply.

The results of experimental modal analysis reflected these properties, with the mean damping ratios of the first ten modes of resonance being inversely proportional to the modulus of the reinforcing fabrics in pure laminates. For hybrid laminates, negative synergy was observed for the mean damping ratio that most strongly correlated with the stiffness of the outermost ply.

In summary, the DBVI process was found to be viable for producing laminates with relatively low void fraction and general compatibility of dissimilar constituent layers. As expected from prior investigations into other hybrid laminates, the material of the outermost ply dominated mechanical and physical properties of hybrid laminates, which resulted in non-linear deviations from predictions made by the rule of mixtures. All of the hybrid laminates showed a unique balance of mechanical and modal properties compared to their constituent counterparts.

In this investigation, hybridized panels with recycled carbon fiber or natural fiber fabrics possessed moderately lower mechanical strength and stiffness and higher damping ratios than their pure virgin carbon fiber counterparts. While the design of composites for lightweight structures has tended to focus on maximizing the strength- and stiffness-to-weight ratio where it has been cost-effective to do so, these hybrid laminates show that it is also possible to increase the damping coefficient of laminates without excessively sacrificing the specific strength or stiffness of the materials. Such hybrids may, therefore, play an important role in applications where greater acoustic damping is required—for example, in the design of the interior panels of vehicles.

In conclusion, the hybrid laminates tested in this paper may find uses in applications where increased vibrational damping performance with a relatively small decrease in mechanical performance is deemed as acceptable, allowing for further routes for the uptake of sustainable reinforcements.

## Figures and Tables

**Figure 1 materials-13-04955-f001:**
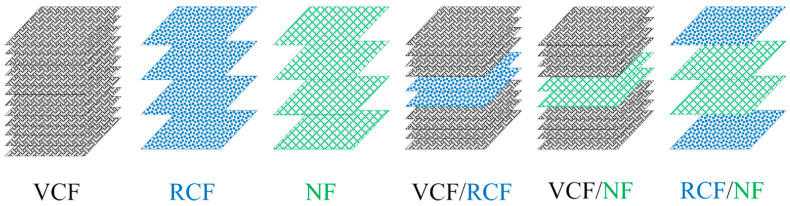
Stacking sequence of pure and hybrid laminates.

**Figure 2 materials-13-04955-f002:**
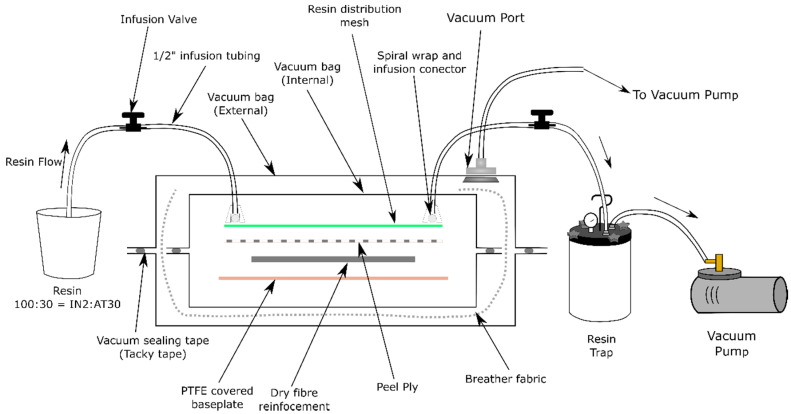
Diagram of double bag vacuum infusion (DBVI) set up.

**Figure 3 materials-13-04955-f003:**
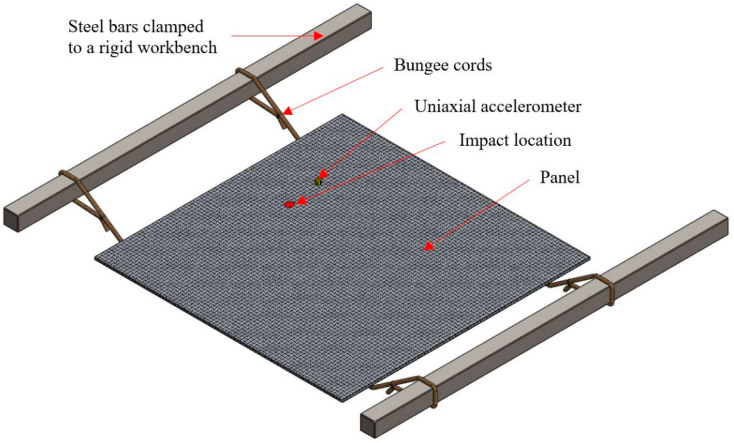
Suspension system for impulse excitation testing.

**Figure 4 materials-13-04955-f004:**
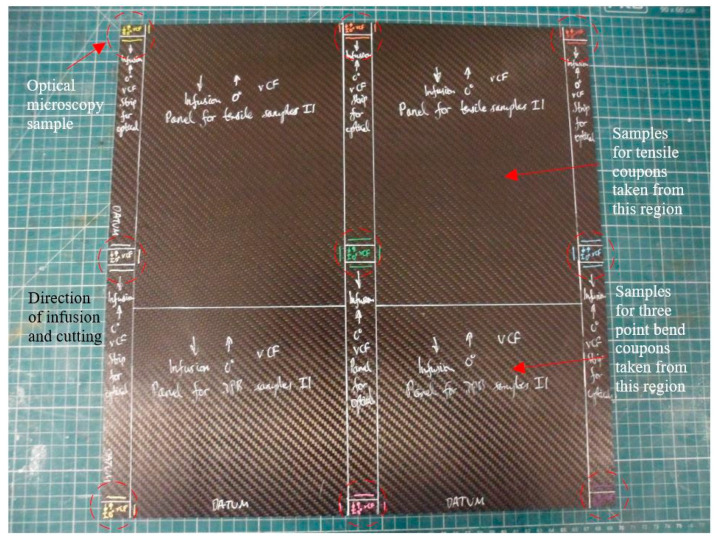
Virgin carbon fiber panel prepared for cutting into coupons and optical microscopy samples following the completion of experimental modal analysis.

**Figure 5 materials-13-04955-f005:**
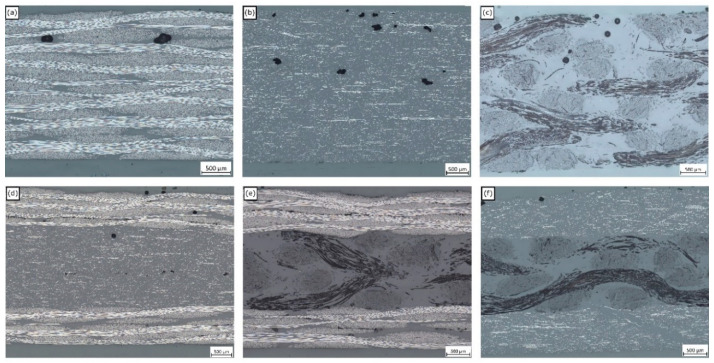
Optical micrographs of representative cross sections from the (**a**) virgin carbon fiber (VCF), (**b**) recycled carbon fiber (RCF), (**c**) natural fiber (NF), (**d**) VCF-RCF, (**e**) VCF-NF and (**f**) RCF-NF laminates.

**Figure 6 materials-13-04955-f006:**
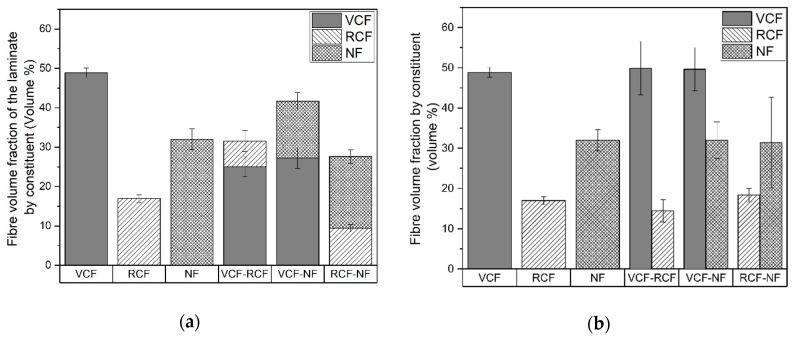
(**a**) Fiber volume fraction of the laminate by the fiber volume fraction of its constituent layers. (**b**) Fiber volume fraction of each constituent layer.

**Figure 7 materials-13-04955-f007:**
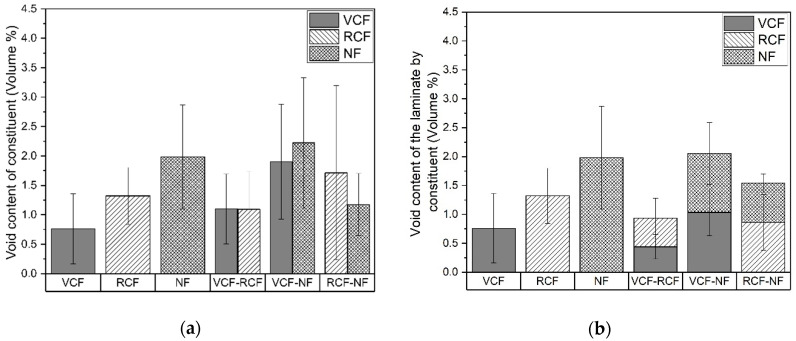
(**a**) Void content of each laminate with respect to the void content of its constituent layers. (**b**) Void content of each laminate as a proportion of its constituent layers.

**Figure 8 materials-13-04955-f008:**
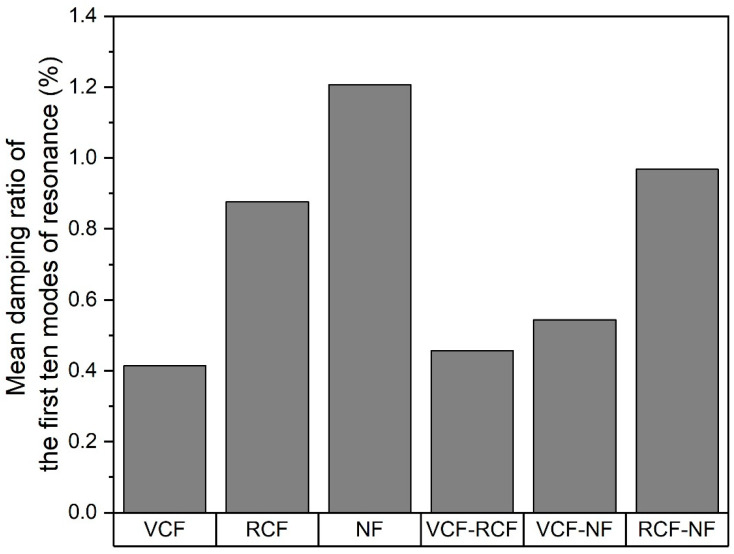
Mean damping ratio of the first ten modes of resonance by reinforcement material.

**Figure 9 materials-13-04955-f009:**
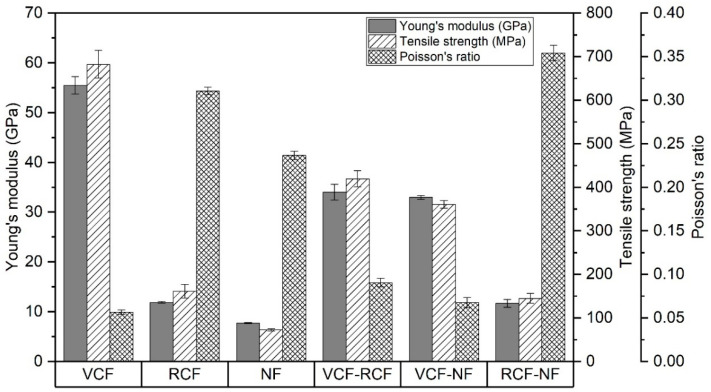
Young’s modulus, tensile strength and Poisson’s ratio results for the pure and hybrid laminates.

**Figure 10 materials-13-04955-f010:**
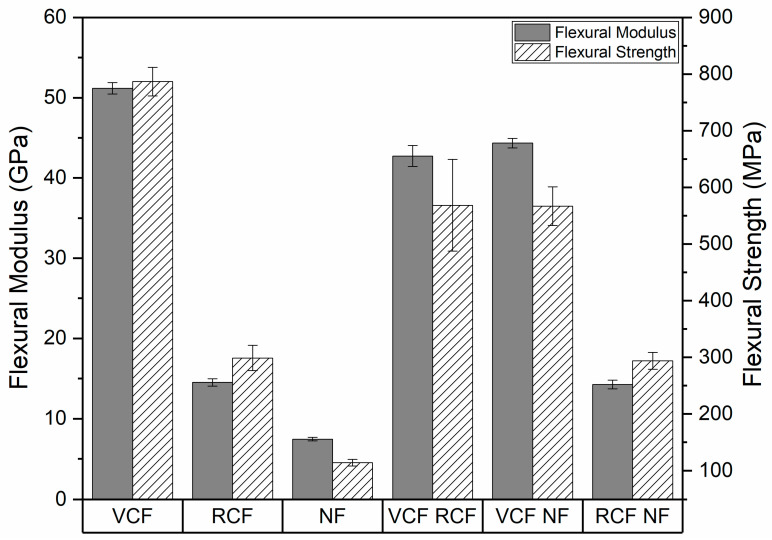
Flexural modulus and flexural strength of the pure and hybrid laminates.

**Table 1 materials-13-04955-t001:** Summary of fiber reinforced plastic panels.

Type	Reinforcement	Designated	Lay-up Sequence (degrees)	Mass (kg)	Thickness (mm)
Pure	Toray T300 carbon fiber	VCF	[(0/90)VCF]_3_s	0.685	2.88
	Carbiso M SM45D carbon fiber	RCF	[(0/90)RCF]s	0.773	3.45
	Biotex flax fiber	NF	[(0/90)NF]s	0.712	3.24
Hybrid	Toray T300 carbon fiber/Carbiso M SM45D carbon fiber	VCF-RCF	[(0/90/90/0)VCF(0)RCF]s	0.874	3.69
	Toray T300 carbon fiber/Biotex flax fiber	VCF-NF	[(0/90/90/0)VCF(0)NF]s	0.715	3.46
	Carbiso M SM45D carbon fiber/Biotex flax fiber	RCF-NF	[(0)RCF(0)NF]s	0.814	3.31

**Table 2 materials-13-04955-t002:** Digital image correlation (DIC) instrumentation and setup parameters.

Parameter	
Camera	Teledyne DALSA 12 M
Lens	Titanar 75 b
Camera angle/°	24.47
Frame rate/Hz	1
Working distance/mm	533
Pixel to mm	1 pixel = 0.03 mm
Aperture	f/22
Imaging window/pixels	4096 × 3000
Calibration Plate	CP20/90/D08713
Measurement volume/mm^2^	125 mm × 95 mm × 75 mm
Facet size/pixels	19
Step size/pixels	13
Spatial resolution/mm	0.57
Displacement resolution/µm	1

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
