# Peer review of "Interlayer Hybridization of Virgin Carbon, Recycled Carbon and Natural Fiber Laminates"

_materials, 2020, doi:10.3390/ma13214955_

Round 1

Reviewer 1 Report

In this paper damping, tensile and flexural properties of hybridized laminates, containing two reinforcements from virgin woven carbon fiber, recycled non-woven carbon fiber and natural fibers were investigated.

The critical point in this paper is due to the different thickness of the laminates manufactured by infusion process. The values of thickness are not reported in table 1, but considering the stacking sequence is possible to suppose value in thickness higher for virgin woven carbon fiberVCF.

In particular for the hybrid laminates with VCF 8 layers were used, while only two for RCFand NF. Laminates constituted by a higher  number of  layers are characterized by a higher volumetric percent of reinforcement.

In the comparison of the mechanical properties , the influence of different values in volumetric percent of  fibres has a fundamental role .

The authors absolutely have to justify the different number of layers used in the  manufacturing of laminates.

Author Response

  • In this paper damping, tensile and flexural properties of hybridized laminates, containing two reinforcements from virgin woven carbon fiber, recycled non-woven carbon fiber and natural fibers were investigated.
  • The critical point in this paper is due to the different thickness of the laminates manufactured by infusion process. The values of thickness are not reported in table 1, but considering the stacking sequence is possible to suppose value in thickness higher for virgin woven carbon fiber VCF.
    • We apologise for the oversight and we have added the missing information the table.
  • In particular for the hybrid laminates with VCF 8 layers were used, while only two for RCF and NF. Laminates constituted by a higher number of layers are characterized by a higher volumetric percent of reinforcement.
    • I appreciate the confusion that has been caused by not having sufficient information about the cured ply thickness. We have addressed this with some amendments to the text.
  • In the comparison of the mechanical properties, the influence of different values in volumetric percent of fibres has a fundamental role.
    • Please see 3.1 for a discussion of the fibre volume fraction of the different panels.
  • The authors absolutely have to justify the different number of layers used in the manufacturing of laminates.
    • Please see the note in 2.1 about variation in cured panel thickness arising from the dissimilarity of available fabrics and efforts taken to choose lay-up sequences that would give balanced panels.

Reviewer 2 Report

This manuscript proposed a sustainable hybrid laminate formed by natural fiber (NF) ,recycled carbon fiber (RCF) and virgin carbon fibers (VCF) using the double bag vacuum infusion (DBVI) method.The mechanical properties and damping ratio of the six materials were compared through experiment.

The study of the hybridization of fiber and recycled carbon fiber is not completely new, however,the attempt to combine the two is worthy of attention. Following are some comments:

1) The results should be analyzed in more depth, for example, how does the average fiber volume fraction affect the damping ratio?

2) How is the contact position selected in Figure 3? What is the basis for selection?

3) Please consider the impact of the Uniaxial accelerometer on the experimental results.

4) How to deal with the void between the laminates in the hybrid laminate?

5)Change "vCF" etc. to "VCF" in Figure 10.

6) Please compare the hybrid laminate with the current materials used in the transport sector to explore its practical significance.

Author Response

This manuscript proposed a sustainable hybrid laminate formed by natural fiber (NF) ,recycled carbon fiber (RCF) and virgin carbon fibers (VCF) using the double bag vacuum infusion (DBVI) method.The mechanical properties and damping ratio of the six materials were compared through experiment.

The study of the hybridization of fiber and recycled carbon fiber is not completely new, however,the attempt to combine the two is worthy of attention. Following are some comments:

1) The results should be analyzed in more depth, for example, how does the average fiber volume fraction affect the damping ratio?

  • The damping ratio is inversely proportional to the stiffness of the panels, which is greater for panels with a higher fibre volume fraction of stiffer reinforcing fibres. Please see this clarification in 3.2.

2) How is the contact position selected in Figure 3? What is the basis for selection?

  • Please see the additional clarification in 2.3 about the methodology for experimental modal analysis.

3) Please consider the impact of the Uniaxial accelerometer on the experimental results.

  • Please see the additional clarification in 2.3 about the methodology for experimental modal analysis.

4) How to deal with the void between the laminates in the hybrid laminate?

  • The presence of a resin-rich region between laminae in hybrid laminates was noted. A section describing its significance for potential applications of the novel material system has been added to 3.1.
  • I’m not sure what Reviewer is asking for in this comment. Do they mean that we should comment about whether or not this is a good thing for the material with respect to possible future applications? We could speculate that resin-rich regions might add parasitic weight but could offer better impact resistance, which is something that could be investigated for further work.

5)Change "vCF" etc. to "VCF" in Figure 10.

  • Thank you for noticing this, Figure 10 corrected

6) Please compare the hybrid laminate with the current materials used in the transport sector to explore its practical significance.

  • Please see the expanded discussion about this in the conclusion in 4.

Reviewer 3 Report

Review report

Manuscript ID: materials-949211

Title:  Interlayer hybridization of virgin carbon, recycled carbon and natural fiber laminates

In this article, the authors have fabricated hybrid laminates using virgin carbon fibre, recycled carbon fibre and natural fibre via double bag vacuum infusion technique. The effect of interlayer hybridization on the mechanical performance of those laminates has been studied. Overall, the manuscript is very well written, and results are well explained. However, there are some minor points which need to be clarified:

  1. How did you determine fibre volume fraction (Vf)? Does this affect the DBVI process? Why %Vf differs in figure 6a and b for hybrid laminates?

  1. Page 11, section 3.3.1, line 299-300: “VCR-RCF and VCF-NF decreased in Young’s modulus by 38.7 % and 40.5 % with respect to pure VCF”- Please check the sample code.

  1. Tensile and flexural strength have been decreased in case of hybrid fibre reinforced laminates compared to neat fibre reinforced laminates. Could incompatibility between hydrophilic and hydrophobic system contribute to this?

Author Response

In this article, the authors have fabricated hybrid laminates using virgin carbon fibre, recycled carbon fibre and natural fibre via double bag vacuum infusion technique. The effect of interlayer hybridization on the mechanical performance of those laminates has been studied. Overall, the manuscript is very well written, and results are well explained. However, there are some minor points which need to be clarified:

  1. How did you determine fibre volume fraction (Vf)? Does this affect the DBVI process? Why %Vf differs in figure 6a and b for hybrid laminates?
    • The method for calculating the fibre volume fraction can be found in the section on optical microscopy in 2.5. This technique was undertaken on samples taken from the panels after they had been manufactured, as stated in the text. As it was not undertaken during manufacturing, it did not interfere with the DBVI process.
    • I have added text to the captions of Figure 6 and Figure 7 in order to remove ambiguity about the meaning of the graphs.
  2. Page 11, section 3.3.1, line 299-300: “VCR-RCF and VCF-NF decreased in Young’s modulus by 38.7 % and 40.5 % with respect to pure VCF”- Please check the sample code.
    • Thank you for noticing this, I have corrected this mistake.
  3. Tensile and flexural strength have been decreased in case of hybrid fibre reinforced laminates compared to neat fibre reinforced laminates. Could incompatibility between hydrophilic and hydrophobic system contribute to this?
    • It is certainly possible that it could contribute. However, we don't believe this is the dominating factor as it would be expected that the hybrids including NF reinforcement would both experience a drop in tensile strength, which is not observed. A rationale for this has been added in section 3.3.1

Round 2

Reviewer 1 Report

Please check in line245-288-329-334: "in Error! Reference source not found.."